# Enhancing Water Safety: Exploring Recent Technological Approaches for Drowning Detection

**DOI:** 10.3390/s24020331

**Published:** 2024-01-05

**Authors:** Salman Jalalifar, Andrew Belford, Eila Erfani, Amir Razmjou, Rouzbeh Abbassi, Masoud Mohseni-Dargah, Mohsen Asadnia

**Affiliations:** 1School of Engineering, Macquarie University, Sydney, NSW 2109, Australia; salman.jalalifar@mq.edu.au (S.J.); andrew.belford@mq.edu.au (A.B.); rouzbeh.abbassi@mq.edu.au (R.A.); masoud.mohsenidargah@students.mq.edu.au (M.M.-D.); 2School of Information Systems and Technology Management, University of New South Wales, Sydney, NSW 1466, Australia; e.erfani@unsw.edu.au; 3School of Engineering, Edith Cowan University, Perth, WA 6027, Australia; amir.razmjou@ecu.edu.au

**Keywords:** drowning detection systems, image processing, sensor-based technologies, water safety, surveillance systems

## Abstract

Drowning poses a significant threat, resulting in unexpected injuries and fatalities. To promote water sports activities, it is crucial to develop surveillance systems that enhance safety around pools and waterways. This paper presents an overview of recent advancements in drowning detection, with a specific focus on image processing and sensor-based methods. Furthermore, the potential of artificial intelligence (AI), machine learning algorithms (MLAs), and robotics technology in this field is explored. The review examines the technological challenges, benefits, and drawbacks associated with these approaches. The findings reveal that image processing and sensor-based technologies are the most effective approaches for drowning detection systems. However, the image-processing approach requires substantial resources and sophisticated MLAs, making it costly and complex to implement. Conversely, sensor-based approaches offer practical, cost-effective, and widely applicable solutions for drowning detection. These approaches involve data transmission from the swimmer’s condition to the processing unit through sensing technology, utilising both wired and wireless communication channels. This paper explores the recent developments in drowning detection systems while considering costs, complexity, and practicality in selecting and implementing such systems. The assessment of various technological approaches contributes to ongoing efforts aimed at improving water safety and reducing the risks associated with drowning incidents.

## 1. Introduction

In recent decades, the popularity of swimming pools and other water attractions has significantly increased. To ensure the safety of individuals in pools, rivers, and beaches, it is crucial to have accessible and well-maintained surveillance systems. Drowning is responsible for approximately 7% of all injury deaths and is ranked the third leading cause of accidental-injury-related death by the World Health Organisation (WHO, Geneva, Switzerland) [1]. Globally, an estimated 230,000 people lose their lives to drowning each year, making it a threat to individuals of all age groups, from infants to senior citizens [2].

Despite the significant risk that drowning poses to public health, global estimation might significantly understate its true extent [3,4]. Specific demographics, such as children and young men, who engage in water-based activities are more susceptible to this danger [5]. Drowning is a multifaceted issue with various underlying causes [6,7,8,9]. However, the primary factors contributing to drowning include the inability to swim, fear of water, and insufficient supervision of children [10]. It is important to note that drowning is often a silent event, and victims rarely exhibit convulsive movements. Instead, they exert considerable energy to keep their heads above water, often unable to call for help or signal their distress. When water enters their larynx or trachea, panic and spasms can occur, preventing them from shouting for assistance. Detecting drowning is typically challenging, as victims may struggle in the water, have trouble breathing or have an irregular heartbeat. Without the intervention of a trained lifeguard, individuals may remain on the water’s surface for only 20 to 60 s before submersion occurs. Therefore, locating a missing person in the water is crucial for their survival.

In recent years, significant progress has been made in advancing technologies that enhance public safety in water-related environments [11,12]. These advancements have brought forth many methods and tools, ranging from simple yet effective techniques to sophisticated systems, all aimed at mitigating risks associated with water-related incidents, particularly drowning. This paper investigates and evaluates various approaches to address this critical issue.

The primary objective of this study is to explore different strategies and assess their performance, cost-effectiveness, complexity, and accuracy. By thoroughly examining these factors, we aim to provide valuable insights into the most efficient and reliable methods for detecting and preventing drowning incidents. Given the time-sensitive nature of such emergencies, the swift identification and analysis of relevant parameters become imperative. The ability to quickly recognise and understand these parameters within a limited time window is crucial for effective intervention and ultimately saving lives. This paper comprehensively reviews strategies encompassing image processing, machine learning implementations, and sensor techniques. These diverse approaches offer unique advantages and have the potential to contribute significantly to drowning detection and prevention efforts. By evaluating the strengths and limitations of each method, we can shed light on their practicality and effectiveness in real-world scenarios.

The World Health Organisation has highlighted that 90% of drowning incidents occur in low- and middle-income countries [1]. Therefore, the cost-effectiveness and ease of access to drowning prevention technologies are crucial factors in determining their applicability in these regions compared to high-income countries. For example, while image processing techniques are efficient, they demand significant investment and complex machine learning algorithms, which makes them expensive and intricate. On the other hand, sensor-based methods offer a more accessible and affordable option for drowning detection. These methods utilise sensors to transmit data about the swimmer’s condition to a central processing unit, employing wired and wireless communication channels.

The analyses by Peden and McGee [13] and Tyler et al. [14] highlight the prevalence and risk factors of drowning in low- and middle-income countries. Peden and McGee’s [13] examination of WHO Global Burden of Disease data reveals that a majority of fatal and non-fatal drowning incidents occur in these regions, with exceptionally high rates in Africa, the Western Pacific Region, and the Southeast Asia Region. Complementing this, Tyler et al.’s [14] review of 62 relevant articles, predominantly from Asia (56%) and Africa (26%), identifies several key risk factors for drowning. These include being young (under 17–20 years old), male gender (75% male vs. 25% female), residing in rural areas (84% rural vs. 16% urban), incidents occurring more frequently during the daytime (95% day vs. 5% night), lack of adult supervision (76% unsupervised vs. 18% supervised), and limited swimming ability (86% non-swimmers vs. 10% swimmers). Additionally, the risk of drowning was similar in both small (42% in ponds, ditches, streams, and wells) and large bodies of water (46% in lakes, rivers, seas, and oceans).

This paper aims to provide researchers, practitioners, and decision-makers with valuable insights into the most promising and viable approaches to enhance water safety by thoroughly examining the performance, cost, complexity, and accuracy of drowning detection and prevention strategies. Ultimately, our collective efforts in understanding and implementing these innovative technologies can significantly reduce drowning incidents and ensure a safer environment for individuals in water-related settings.

## 2. Drowning Behaviour, Signs, and Statistics

From 2008 to 2020, Connecticut researchers [15] conducted autopsies on 500 bodies retrieved from the water categorised by age, gender, location, type of water body, cause of death, method of death, decomposition symptoms, body mass index (BMI), brain weight, lung weight, pulmonary oedema, stomach contents, and toxicological tests. Men more frequently drown than women (excluding cases of suicide). Two common characteristics of drowning deaths are water retention in the lungs and brain swelling. It is essential to look at the big picture while assessing anatomical data. In salt water, the brain and lungs both increase in mass. However, BMI and fat distribution also have a role. Figure 1a shows the total number of drowning victims at different locations. As depicted in Figure 1b, swimming or other recreational activities are the most common activities leading to drowning, regardless of age or location (except in the ocean or a harbour). In 80% of drowning deaths, it was unclear if drugs or alcohol played a role. However, among the remaining 20% where such involvement was determined, 58% involved legal drugs (56% involved alcohol), 19% involved illegal drugs, 15% involved legal and illegal drugs, and 8% did not involve either [16].

Drowning can be categorised as active or passive. Active drowning is when a person actively tries to swim to the surface. For example, the swimmer can perform “ladder climbing” or other actions that cause them to break the water’s surface (splashing and flailing their limbs), or they may remain submerged without generating an audible sound or visual ripple. If a swimmer is actively drowning, they are either immersed or unable to keep their head above water and cannot call or signal for help. Passive drowning occurs when swimmers lose consciousness while submerged in water, preventing them from resisting and leading to death [17,18]. It was commonly thought that non-swimmers drown due to a ‘fight or flight’ response characterized by frantic movements. However, observations from witnesses indicate that drowning typically does not involve such thrashing behaviours. In some instances, a body was reported to float on the water when it stopped moving. The swimmer sank and disappeared. These reports show that drowning is a hidden killer, as the victim cannot signal for help.

While drowning mainly results from asphyxiation, incidents involving falls or dives into the ocean can sometimes lead to secondary complications such as spinal or brain injuries, which are exacerbated by oxygen deprivation but are not direct causes of drowning. Alcohol and drugs can also influence drowning and death. Drowning is a form of death through asphyxiation and severe cerebral hypoxia, also known as suffocation [19,20]. Both active and passive drowning have phases that last 10–12 min. During this interval, the person can be rescued and resuscitated without permanent harm: (1) The person inhales water. (2) If a person is at risk of drowning or oxygen depletion, their airways will close. (3) The swimmer loses consciousness, and their body slows down to conserve oxygen. (4) Water enters the lungs through the windpipe. (5) Low oxygen causes muscle twitching. (6) Without oxygen, brain damage is permanent. (7) Oxygen-deprived brain death occurs. Before stage five, a drowning swimmer can be rescued and resuscitated without long-term health effects. However, once brain damage from lack of oxygen commences, the chances of rescue and survival without long-term implications quickly diminish until stage seven. Therefore, while it takes seconds for someone to start drowning, it takes minutes to accomplish a rescue without irreversible damage, and any drowning prevention method must account for this [18,21].

Drowning often results in death and injury, but in fortunate cases, the aftermath of a non-fatal drowning may not result in significant health issues. Nonetheless, there have been documented instances where non-fatal drowning has negatively affected a person’s health and quality of life [22,23,24,25,26,27]. The brain or other organs may sustain damage, a condition known as hypoxic brain injury (owing to oxygen shortage). Preventable drowning is the leading cause of death for children under four. It has been found that swimming pools are the most common location for non-fatal drownings, especially among children [28]. In contrast, most adult and adolescent drownings occur in natural bodies of water such as lakes, rivers, and beaches [29].

The National Drowning Report by the Royal Life Saving Society Australia for 2022/23 [16] comprehensively analyses drowning incidents in Australia. It revealed a marginal increase in drownings, with 281 cases, a 1% rise compared to the 10-year average of 279. A significant 77% of these incidents involved males. Adults aged 45 and older accounted for 57% of the drownings, with 44% occurring in major cities. Rivers and creeks, beaches, and oceans/harbours were the most common locations, accounting for 27%, 27%, and 12% of incidents, respectively. The primary activities leading to drownings were swimming and recreating (33%), falling into water (15%), and boating (8%). Children aged 0–4 years and adults over 75 years were particularly vulnerable to drowning due to falls into water, representing 69% and 22% of their respective age group drownings. Notably, boating-related drownings decreased by 40% compared to the decade-long average. For young children under four years, fatal drowning cases decreased significantly, representing 6% of the total in 2022/23, marking a 6% drop from the previous year and a 33% decline from the 10-year average. Similarly, the crude fatal drowning rate for this age group decreased by 32% from the 10-year average and 59% from the rates observed 20 years ago. Drownings in the 5–14-year age group constituted 2% of the total, a 53% reduction from the previous year and a 35% decrease from the 10-year average. The 15–24-year age group represented 11% of drowning cases, down 17% from the previous year and 7% from the 10-year average. Interestingly, the 25–64-year age group, which accounted for 54% of drowning deaths, showed a 1% increase compared to the 10-year average. The age groups with the highest drowning rates were 45–54 years, 55–64 years, and 65–74 years, representing 15%, 15%, and 14% of the drownings, respectively. This is important when training AI models. If it is not trained using images of sufficient numbers of both males and females, an algorithm may achieve, for instance, 80% accuracy, while failing to safeguard any females. While the report does not explicitly address race, it is important to consider factors like age, gender, and other visible differences to ensure that any model appropriately protects everyone who could use or purchase an AI-based drowning detection device. Table 1 provides a summary of the most common technologies for drowning detection.

Although the number of drowning deaths is important, it is not the whole story. For every fatal drowning, there are around 2.5–2.7 times as many non-fatal episodes [22]. Furthermore, drowning can result in various adverse consequences other than death. Only 5% of people who survive drowning do so without suffering a permanent handicap. Considering the years of life lost, estimated lost productivity, and hospitalisation and search and rescue expenses, the average annual economic impact of fatal drownings in Australia was calculated to be around 1.24 billion AUD for the period from 2002 to 2017 [30]. Contrary to popular belief, drowning is extremely difficult to detect unless one is trained to detect it. There are a few basic movements or physical indications which usually indicate a swimmer is in distress, such as (1) agitated movement of the arms, (2) glossy or closed eyes, (3) tilted head with the mouth barely keeping above water level, (4) hair covering the entire face/forehead, thus obscuring a person’s vision, (5) aberrant breathing and hyperventilating, (6) swimming without a direction, and (7) floating on one’s back without any leg movement [31]. These signs may be commonplace indicators of drowning, but only a few are measurable parameters and are consistent occurrences for every scenario. Research shows this is mainly because drowning indicators for one person may not be applicable or legitimate for another. This can lead to either false alarms or, in some cases, failure of detection [32].

## 3. Image Processing Techniques for Drowning Detection

Many drowning incidents occur in places without adequate supervision. It is prohibitively costly to have lifeguards monitoring large portions of the coastline. Thus, they are mainly tasked to patrol popular beaches. Additionally, lifeguards do not patrol most rivers or residential pools. Due to the expense, comprehensive monitoring is not feasible. The alternative is to develop a low-cost autonomous system that identifies swimmers in distress from an above-water vantage point, such as a pole, observation tower, or a mobile Unmanned Aerial Vehicle (UAV), such as a quadcopter or fixed-wing aircraft. Previous research into this field has focused on the use of autonomous systems to detect swimmers (regardless of their distress level) [33] or UAVs to capture better footage for trained employees to monitor [34].

Joseph Redmon [35] designed Darknet for neural image recognition. It is being updated by AlexeyAB and is used for object and human recognition. Redmon’s You Only Look Once (YOLO) model is one of the most well known and is the basis of many modern research articles on this issue. Many writers have released improved versions of YOLO using Darknet as a starting point. MS COCO is an extensive library of pre-labelled images with 80 classes and daily objects. Mean average precision (mAP), a measure of “overlap between the prediction and ground truth bounding boxes” and predicted class accuracy, is used to assess model performance. Speed can be expressed as a framerate by finding the inverse of the time it takes to perform inference (process a single frame) [36]. Some networks can add processing steps to increase application performance. For example, Recurrent Only Look Once (ROLO) adds a long short-term memory (LSTM) layer to the YOLO pipeline to provide recurrent neural network (RNN) capability, boosting the model’s performance on occluded objects [37]. If a CNN or YOLO model has been trained, any picture may be used, including thermal or other camera technologies that can identify humans [38,39]. In addition, they are fast enough to run on mobile phones, computers, and various platforms.

Studies have successfully distinguished people on land from those swimming, using trained vision systems [40]. Typically, search and rescue (SAR) applications motivate research into detecting swimmers, such as autonomously locating lost people at great distances [33]. It is crucial to detect and differentiate between swimmers who are drowning and others who may be in the frame but are not swimmers, e.g., those on the shoreline or poolside. Drowning detection has been studied for swimmers at sea [41] and in pools [42]. From an engineering standpoint, pools provide a convenient development platform. Above-water cameras work well in pools with high-contrast walls. These walls contrast nicely with all skin tones, making detection easier. In addition, pool locations facilitate underwater cameras and provide mounting locations for this equipment. In contrast, an ocean environment has a sandy bottom, and the water is only sometimes clear enough for them to perform effectively due to sediments such as sand and marine life, which reduce visibility to an unusable level [42,43].

Drowning can be detected through video surveillance in three ways, by employing (a) submerged cameras to monitor whether the swimmer’s body is sunken, (b) a stationary camera to detect the location of forceful splashes on the water, and a spinning underwater camera to assess the location of the splashes by submerged individuals, and (c) a camera that distinguishes only the drowning characteristics. Rooz et al. [44] were the first to suggest using sonar. Active sonar is used to scan the pool area, and the technology can distinguish between people and inanimate objects in the captured photos. An underwater camera was then proposed as an alternative method [45,46] for detecting people near the edge of the pool. However, this strategy largely overshadowed camera-based techniques. Both solutions are costly and inconvenient due to underwater installation and housing. In addition, underwater-camera-based systems suffer from blind spots. Existing drowning detection approaches mimic drowning characteristics for supervised classification, but drowning events are infrequent and difficult to simulate. Also, unexpected actions or appearances cause anomalies in video footage. There are several methods for finding video anomalies, but the most important step is extracting video features [47,48,49,50]. Video anomaly detection tasks are another area where deep learning algorithms have had recent success [51,52,53]. Since high-level semantic aspects of films can be extracted using deep learning algorithms, together with the spatial and temporal elements, they are also more resilient in dealing with complicated scenes.

Feature extraction, feature expression, and evaluation criteria are available for estimating typical swimming speed based on swimmers’ motion capture—determining orientation independent of the camera angle and assessing a risky situation, as depicted by the chart in Figure 2 [54]. An alarm is triggered if no head is located within a given time range. If no bounding box overlaps the initial bounding box of the head, then the target is not swimming. In that case, whether the swimmer is treading water upright is assessed. An alert will sound if the swimmer is in a risky upright posture. The typical swimming speed may be calculated if the number of sample frames and intervals is known. This approach helps identify any changes in the swimmer’s speed and determine if these changes are unusual. When the threshold is exceeded, an assessment of a dangerous upright condition is performed. Again, a warning will sound if the swimmer is upright and in a dangerous posture. Using the bounding box of the swimmer’s head and the overlapping area of the neighbouring bounding box eliminates the error created by directly applying the object displacement time ratio. When the camera’s optical axis is not perpendicular to the swimmer’s direction, the long and short axes of the outer ellipse of the human body do not operate. However, in such scenarios, measuring the ratio of the head size to the body size provides a more accurate way to determine if the swimmer is in an upright position.

As shown in Figure 3, the approach for swimmer status extraction comprises a learning phase that constructs the initial background model and a detection phase that separates swimmers from the non-stationary pool background [42]. Addressing the issues experienced by automated surveillance systems under challenging contexts is necessary for developing algorithms for a system capable of detecting water emergencies in highly dynamic aquatic environments. A robust segmentation algorithm based on block-based background modelling and thresholding with a hysteresis approach enables reliable detection of swimmers in reflections, ripples, splashes, and fast illumination changes. A Markov random field architecture addresses partial occlusions, enhancing the system’s tracking capabilities. The relatively high noise level in the foreground detection and behaviour recognition steps represent a significant obstacle to solving the problem [55]. As a result, background subtraction, denoising, data fusion, and blob-splitting techniques are used, motivated by the peculiarities of the aquatic background and the crowded pool situation. In addition, visual signs of distress and drowning are integrated via a set of foreground descriptors in the detection stage for early drowning. Also, a module consisting of data fusion and hidden Markov modelling is built to discover the distinctive characteristics of various swimming behaviours, notably those early drowning incidents.

The first step in an unsupervised method for identifying drowning incidents in pools is gathering pool images. Incidents involving drowning are only included in the test set, not the training set. After the dataset is prepared, a ResNet-modified neural network is presented for reconstructing video frames [56]. When anomalous events occur, they can be found by comparing the reconstructed frames to the ground truth frames. Regarding the method quality outcomes in the dataset, the water surface region always generates substantial inaccuracies due to water and light variations. The water surface flaws are always present in normal and abnormal frames. Therefore, they cancel each other out when compared and do not influence the detection of abnormal frames. The error map will only show a slight deviation when someone acts following their typical behaviour. Someone drowning will generate abnormal frames, and the related region error grows considerably.

A video surveillance system can detect drowning accidents using advanced person tracking and semantic event detection technology [57]. Despite significant water ripples, splashes, and shadows, successful detection and tracking of swimmers are possible with a background model that efficiently includes updated information on swimming pools and aquatic environments. In the simulation of water distress conducted by a volunteer and the appropriate system reaction [57], activity and splash index with medium and high values are proposed as criteria for drowning identification. For example, an index of 1.7 is considered a medium value, whereas an index of 2 is a high number and a symptom of drowning. Similarly, the difference between the medium splash index of 400 (safe) and the high splash index of 600 (alarm) is another method for detecting drowning. Thus, drowning is noticed when the activity and splash index values exceed their high limits. When underwater cameras capture the input video sequences, background subtraction will differentiate between moving objects and the background within the caution zone [58]. In addition, convoluted water interferences are simplified using an inter-frame-based denoising technique. The inter-frame-based denoising process will not affect the final image quality. To clarify things, the system is set up to label each ROI on the frame with the duration; if the time is longer than the threshold, an extra “D” would appear on the label, designating the associated region as belonging to a drowned swimmer [58].

Robotic technology may assist in reducing drownings in swimming pools. For instance, Laxman and Jain [59] used submerged and buoyancy-dependent lifts, motion detectors, and laser tripwires to create fail-proof autonomous rescue procedures. In order to counteract a number of the shortcomings of submerged cameras and systems [44,45,60], a study by Jose and Udupa [61] examines several approaches for screening swimming pools using the above cameras [62]. Most drowning detection systems consist solely of the detecting component, which a skilled lifeguard can cover. Fear and adrenaline make drowning individuals quite twice as strong as normal, and they will grasp anything to survive. Therefore, only a trained lifeguard can effectively save a drowning person. Using locations from a camera above, the suggested robot design in [61] will provide the correct assistance at the right moment, eliminating the need for lifeguards. The fully automated mechanism can transport the sufferer to the pool edge. As seen in Figure 4, the suggested system comprises three components: (1) an LED display with an alert system, (2) a top-mounted camera, and (3) a gantry robot. The camera is equipped with a drowning-detecting system that examines the shape and velocity of the swimmer. When the camera detects drowning, the system sends coordinates to the gantry. The camera coordinates to tell the robot where to release the ring buoy. A nylon chain and ring buoy will help the distressed swimmer ascend the drift. A load cell at the end can determine if someone climbed on the dropped buoy by measuring its effective weight. If the system detects a load, it pulls the victim on the ring buoy back to the home position. When the camera detects drowning, it activates the LED display and alarm unit to alert rescuers.

## 4. Sensor-Based Techniques for Drowning Detection

Various research efforts have evaluated and implemented sensors to detect drowning. The anti-drowning system created by Chaudhari et al. [63] is an example of this type of study. In their method, heart rate is monitored, compared to a predetermined threshold, and transmitted using radio frequency (RF) with a range of 5–6 m. The device may be attached to the swimmer’s head or hand for movement assessment. Kulkarni et al. [64] devised a system for detecting drowning utilising non-invasive oxygen saturation, respiration, and water sensors [65,66,67,68,69]. These measure blood oxygen saturation, breathing, and submersion. The controllers are used to monitor, process, and control. The system recognises drowning if any of the features mentioned earlier are present. Hemalatha et al. [70] described a similar system with a receiver and transmitter. The wristband-mounted transmitter module includes a microprocessor, RF transmitter, battery, and pressure sensor. This device compares sensor data to a threshold and generates an output depending on the comparison, similar to Chaudhari et al. [63]. John et al. [71] demonstrated a heart rate and pressure sensor module-based system. The wristband design includes a microcontroller, a heart rate sensor, and a blood pressure sensor with high and low thresholds on the transmitter side. Similar to prior two-module systems, a receiver alert is delivered if a heart pressure reading exceeds the threshold. Ramdhan et al. [72] implemented pulse sensors in a three-module sensor-based system. The system is divided into a monitoring system, an access point, and a drowning detection module. Like previous systems, data would be obtained and compared with predetermined values. Researchers have also studied simple sensor systems to deploy airbags for flotation. For example, Ramani et al. [73] presented an automated drowning prevention system utilising a peripheral interface controller (PIC) microcontroller and accelerometer. When threshold readings are exceeded, the wristband opens an airbag to float the wearer to the water surface. An electric pulse triggers the airbag by burning sodium azide pellets in its inflator. Burning creates nitrogen gas, which fills the airbag and lifts the person to the surface. Nagalikitha et al. [74] suggest a similar system using pressure sensors and accelerometers to initiate the inflation of an airbag.

Survey results [75] indicate that if additional parameters are monitored simultaneously, the accuracy of sensor-based approaches can be improved to a level equivalent to other high-accuracy but more expensive methods, such as image-processing techniques. However, more research is still needed to improve the performance of the sensor-based systems. If one sensor in a multi-sensor device does not promptly pick up on the drowning scenario, other sensors will. One example is the waterproof device for detecting drowning presented by Jalalifar et al. [4]. The device includes heart rate, blood oxygen saturation, swimming patterns, and depth sensors. Each sensor can operate autonomously to increase the system’s anti-drowning capabilities. The microcontroller compares collected data from sensors with user-defined criteria to determine an impending threat. The warning will display if the danger persists over a specified interval. A drowning or safe message may be shown by comparing these values and keeping track of time underwater. The data collected by the sensors may be seen on an OLED screen, connected mobile device, or computer. Unfortunately, the device cannot transmit Wi-Fi signals while submerged in water. Figure 5 depicts the proposed sensor-based system, which uses a microprocessor, sensors, a communication system, and a battery.

### 4.1. Measurable Behaviour

The drowning process can take only tens to hundreds of seconds, making rescue operations difficult [18]. Submersion for more than 4–6 min without resuscitation causes brain damage and leads to death. Research at the University of New England shows that a non-fatal drowning heart rate and blood oxygen level should be checked for potential drowning victims [10]. Only those parameters that can operate as consistent, reliable parameters should be considered when deciding which to use for measurement. Based on these and other studies, the following variables were chosen: heart rate [63,71,72], blood oxygen saturation [64,76], depth of immersion [70], pattern of movement [73], and body temperature [77,78]. All the mentioned factors are reliable, with values that may vary from person to person but stay within a stable range. As a result, the value may be adjusted for each user to provide timely drowning warnings.

#### 4.1.1. Heart Rate

Early studies by researchers [79] divided drowning into two groups. Variations in blood pressure and breathing patterns divided the groupings, while heart rate patterns were common to both. In both cases, heart rates fell before death. A low heart rate leads to a decline in blood oxygen saturation due to the diminished capacity of the heart to circulate oxygenated blood into the body. Norwegian researchers [80] analysed the heart rate data of a man who had fallen from his kayak. He was wearing a wristwatch capable of recording heart rate and temperature. The researchers extracted the data and observed that within minutes after submersion, the subject’s heart rate decreased to 43 beats per minute (BPM) and varied between 78 and 35 BPM. In one study [4], 55 BPM was set as the threshold value while testing a device. This implies that drowning is indicated when the heart rate is 55 BPM or lower.

#### 4.1.2. Blood Oxygen Saturation

Oxygen saturation (SpO_2_) is measured as the ratio of the amount of oxygen-carrying haemoglobin to the amount of haemoglobin not carrying oxygen [81]. For instance, the SpO_2_ is calculated using Equation (1) [82,83]:(1)$${\mathrm{SPO}}_{2}=100\times \frac{{C\;[\mathrm{HbO}}_{2}]}{{C\;[\mathrm{HbO}}_{2}]+C\;[\mathrm{RHb}]}$$
where the oxygenated haemoglobin and deoxyhaemoglobin are defined as HbO_2_ and RHb, respectively. Additionally, the concentration of oxygenated haemoglobin and deoxyhaemoglobin is represented as C.

Every individual requires a certain amount of oxygen in their blood to function normally. Inadequate SpO_2_ levels can result in severe symptoms, indicating the onset of hypoxemia. In addition, there is a skin condition known as cyanosis because of its blue (cyan) colouring. Hypoxemia (low oxygen levels in the blood) can lead to hypoxia (low oxygen levels in the tissue) [81]. As all drowning victims suffer from hypoxia, prompt oxygen administration is one of the most important therapies. However, conventional oxygen administration is 100%, which might harm patients during resuscitation [84]. As the oxygen level in the bloodstream is very stable, it is known as an indicator of drowning [4,75]. The usual range of oxygen saturation in the human body is 95–100% [85], and anything below this value is considered hypoxia [81]. Therefore, when a person is drowning, their oxygen levels will drop dramatically, as expected; the quantity of oxygen in a person’s system would effectively predict whether or not they are experiencing asphyxia due to drowning.

#### 4.1.3. Depth

Water depth was one of four parameters studied when the recovery record of 56 thousand rescues in a lifeguarded waterpark was analysed [86]. The study defines *shallow water* as any depth of ≤1.5 m, while deep water is >1.5 m. The study also showed that 42% of all rescues occurred in shallow water, and 56.6% of recoveries occurred in deep water. Although there is no set depth at which drowning can occur, there is no disputing the fact that water depth influences its prevalence. As divers would have a higher threshold than swimmers in a shallow pool, our device must have a depth measurement option while also providing the possibility to adjust the depth thresholds according to the swimmer’s surroundings and level of capability. Thus, it can be inferred that depth would be a crucial parameter to monitor and use in a drowning detector.

#### 4.1.4. Body Temperature

Hypothermia is a medical disorder in which a human body’s heat loss rate is higher than the heat gain, leading to a dangerously low body temperature. The average body temperature is around 35 °C. Hypothermia occurs when the body temperature drops below 35 °C. This can develop with prolonged exposure to water below 20 °C. When the body’s temperature falls below 32 °C, the heart, brain, and other organs cease functioning properly [78,87]. Safety experts say that many drowning fatalities occur due to hypothermia rather than water in the lungs. The cold water removes heat from the body 25 to 30 times quicker than air. Hypothermia occurs when a person loses enough body heat to drop below normal temperature. Immersion in cold water instantly cools the skin and surrounding tissues. Core body temperature (brain, spinal cord, heart, and lungs) declines within 10 to 15 min. The arms and legs become completely numb and unusable. Before the core temperature goes low enough to cause death, the individual may lose consciousness and drown [77].

#### 4.1.5. Time Duration

The consequence of submersion time for drowning fatalities among those in open water bodies was examined in depth by researchers in Washington [88]. The study concluded that swimmers often died when submerged for over 15 min. It is vital to remember that this period encompasses everything from submersion to cardiac arrest. The goal is to stop drowning before it starts rather than trying to save someone about to pass away. A person can be at risk of drowning with the entry of as little as half a cup of water into their lungs, a situation that can occur in less than 60 seconds. A submersion time of approximately 15 seconds is established as the critical threshold for safety purposes. Then, if any of the other signs mentioned earlier are activated within these 15 s without interruption for a continuous 15 s, this would imply drowning, and the corresponding rescue notification would be signalled.

### 4.2. Developed Domestic Sensor-Based Devices for Children

According to the Royal Life Saving Society Australia [16], 75% of toddler deaths under the age of four resulted from drowning, with half of all deaths occurring in swimming pools, followed by bathtubs/spas and lakes/dams. Similar findings have been reported in the United States and Japan [89], indicating that one-to-four-year-old children had the highest incidence of drowning. Most child drownings in this age group occur in swimming pools. However, drowning may occur at any moment, particularly when children should not be near water, such as when they have unsupervised access to pools [90]. For children aged 1 to 14, drowning is the second leading cause of accidental injury mortality, behind motor vehicle accidents. In addition, non-fatal infant drowning can result in long-term health issues, including brain damage and expensive hospital admissions [91]. The neuropsychological result of children who have suffered a non-fatal drowning episode cannot be adequately anticipated early in the therapeutic process. Therefore, vigorous outpatient and inpatient care is essential. Despite considerably decreased submersion periods, many survivors suffer from severe neurological impairment. These conditions will have a lasting and significant impact on both their families and society throughout their lives. [92]. Parental supervision is crucial in preventing drowning, and swimming pools should be fenced, yet children continue to drown. Prevention of drowning requires a tiered strategy. The more “layers of protection” between a child and water, the more negligible the risk. Therefore, early detection of drowning incidents is necessary for this vulnerable age group when roaming near water.

As presented in Table 2, Multiple home-based safety devices have been developed and produced to enhance children’s safety around swimming pools, spas, and bathtubs. While these devices offer valuable assistance in preventing accidents, it is essential to recognise them as components of a comprehensive approach to water safety. This holistic strategy should also encompass consistent adult supervision, implementing physical barriers, and providing thorough water safety education.

## 5. Conclusions

This paper provides a comprehensive understanding of the most favourable methods of drowning detection, image processing, and sensor-based approaches. While expensive and challenging, image processing offers high accuracy and real-time detection, including underwater, but relies on costly AI and infrastructure, limiting its use to specific locations. Sensor-based methods are more affordable and universal, which makes them a viable option for low-to-middle-income countries, benefiting from advancements in machine learning and IoT for customisation. However, they require individual calibration and face challenges in motion tracking standards and underwater communication. Integrating AI and mobile apps can enhance personalisation in sensor-based systems.

To achieve optimal performance with sensor-based methods, it is crucial to customise the device for each user, considering factors such as age, health, and gender. Fortunately, advancements in machine learning algorithms (MLAs) and the Internet of Things (IoT) now enable the practical adaptation of sensor-based devices. Conversely, image processing methods rely on high-end AI techniques and costly infrastructure, limiting their applicability to specific installations, such as pools and designated areas. In contrast, sensor-based methods offer universality in their application. Despite significant technological advancements in drowning detection systems, notable areas still require attention. Sensor-based methods for tracking motion present room for further development, particularly in establishing standardised ranges for acceptable acceleration and velocity thresholds during swimming. Underwater wireless communication in sensor-based approaches also poses challenges that must be addressed.

Nevertheless, integrating removable components that emerge from the water or float on its surface in a drowning incident can effectively alert lifeguards. AI can be leveraged in sensor-based techniques to personalise measurable parameters for individuals, considering their unique physical and health conditions, age, gender, and other factors. Additionally, the development of a mobile app holds the potential to personalise individual parameter settings.

This paper underscores the strengths and weaknesses of different drowning detection approaches, paving the way for further advancements in water safety and preventing drowning incidents. By addressing the identified shortcomings, we can enhance the effectiveness and reliability of sensor-based systems, ensuring improved safety measures for individuals engaged in water activities.

## Figures and Tables

**Figure 1 sensors-24-00331-f001:**
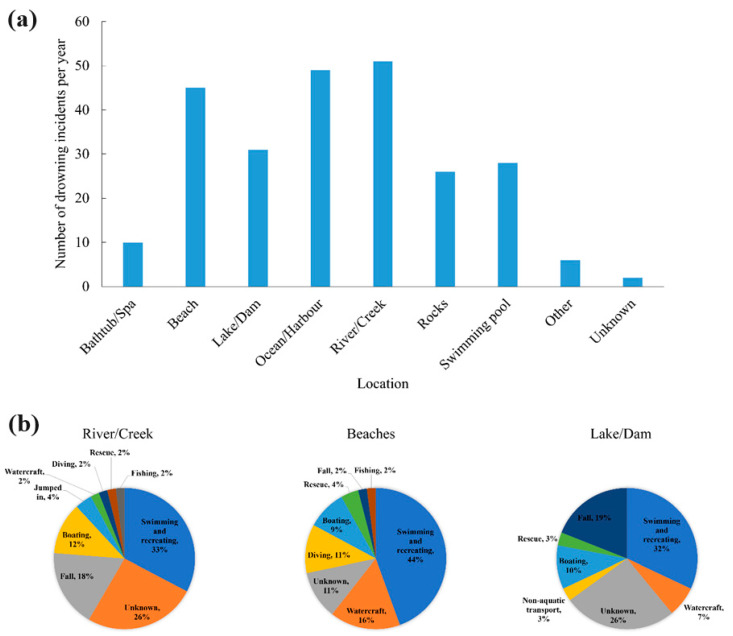
(**a**) Number of yearly drowning deaths by location. (**b**) Activities before drowning incidents by location (river/creek; beaches; lake/dam) in 2019/2022 [16].

**Figure 2 sensors-24-00331-f002:**
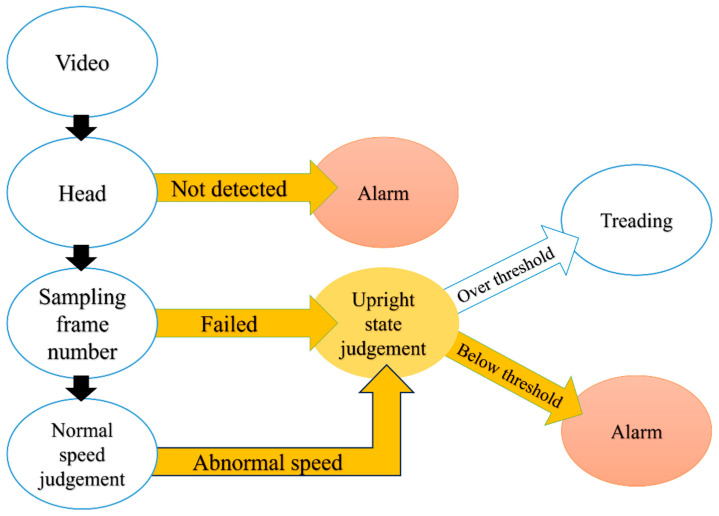
Threat decision of swimmers [54].

**Figure 3 sensors-24-00331-f003:**
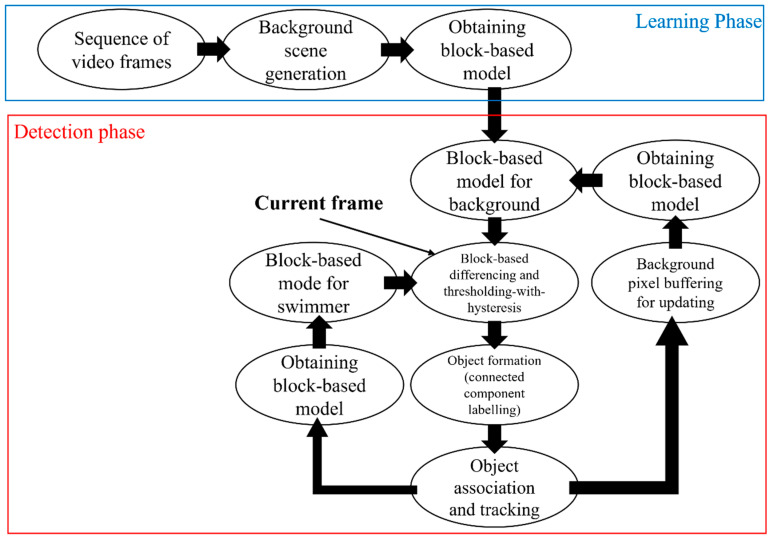
Design of the suggested algorithm for segmentation [42].

**Figure 4 sensors-24-00331-f004:**
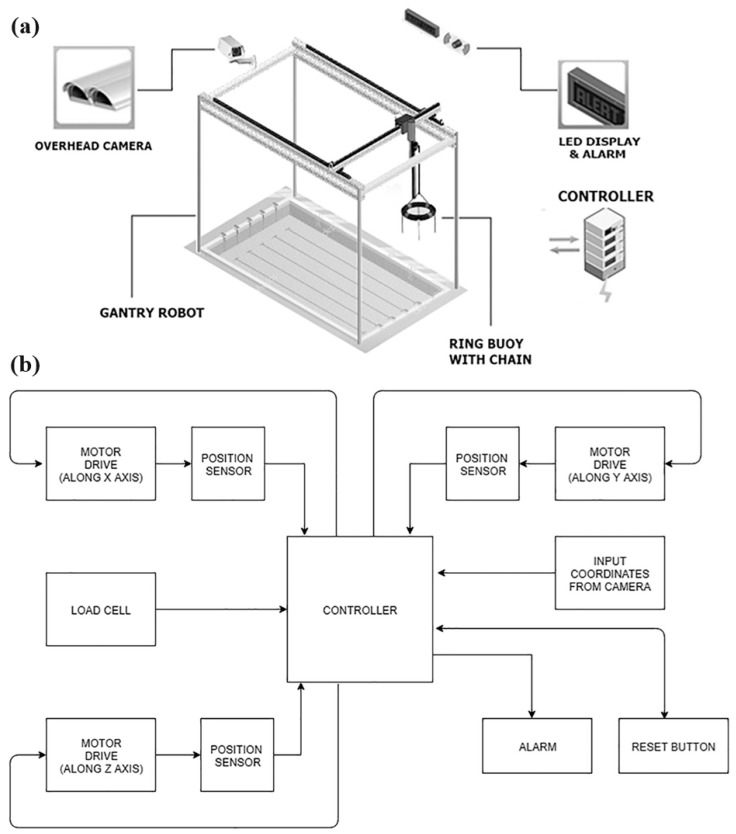
A hybrid camera and overhead gantry robot. (**a**) System description and (**b**) block diagram. Reprinted with permission from [61].

**Figure 5 sensors-24-00331-f005:**
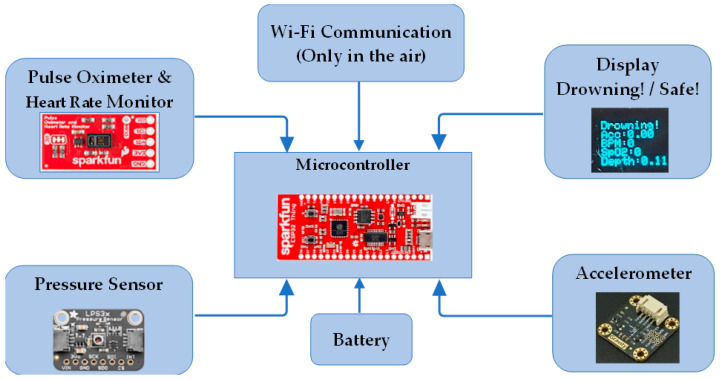
Components of a sensing-based drowning detection system [4].

**Table 1 sensors-24-00331-t001:** Summary of recent methods of drowning detection with advantages and limitations.

Method	Advantages	Limitations
Lifeguard	Human judgment and experience Immediate response and rescue capability Communication and coordination with authorities	Limited physical endurance and potential for fatigue Restricted field of view and potential for human error
Video processing-drones	Aerial view for wide coverage and surveillance Rapid response and quick deployment Real-time video feed for situational awareness	Limited battery life and flight time Restricted flight regulations and airspace limitations Vulnerability to environmental conditions (e.g., wind) Affordability and accessibility in low-to-medium income countries.
Image-processing technology	Continuous monitoring and real-time data collection Accuracy and effectiveness in recognising and differentiating drowning incidents. Customisable thresholds for personalised detection Underwater drowning detection	Limited detection range and requires regular maintenance and calibration. Expensive and complex to implement. Requiring high-end AI techniques and infrastructure. Affordability and accessibility in low-to-medium income countries. Dependence on camera placement and quality
Wearable sensors	Continuous monitoring and real-time data collection Cost-effectiveness and wide availability Customisable thresholds for personalised detection Easy integration with artificial intelligence system which increases the accuracy. Affordability and accessibility in low-to-medium income countries.	Sensitivity to environmental conditions and false alarms Limited detection range and coverage Maintenance and calibration requirements Limited underwater communication

**Table 2 sensors-24-00331-t002:** Overview of functionality and key limitations of child safety devices for home pools and baths.

Device	Functionality	Limitations
Triaxial accelerometer device for bathtub [89]	Monitor water wave motion in a bathtub using a triaxial accelerometer. Identify aberrant wave motion. Alerts are sent to guardians through radio transmission.	**Limited to Bathtubs**: Specifically designed for bathtubs, its effectiveness in larger bodies of water like swimming pools needs to be clarified. **Dependence on Wave Motion Detection**: It may only detect drowning if the water wave motion is sufficiently aberrant. **Radio Transmission Range**: The effectiveness of alerts depends on the range and reliability of the radio transmission.
WiEyeTNB [93]	Designed for use in wireless networks at homes or schools. The water-level detection sensor activates upon contact with water above a predetermined level. A small, wearable earring sensor is positioned at the same level as a person’s mouth. It includes a central processing unit comprising a low-power microprocessor with modest storage (on-chip and flash memory). A ZigBee transceiver is used for low-cost wireless communication within the unlicensed RF Industrial, Scientific, and Medical (ISM) bands (Range: 100 m). Powered by a lightweight, rechargeable, or non-rechargeable battery.	**Water-Level Detection Limitation**: This relies on water reaching a certain level; it may not detect if a child enters the water without significantly raising the level. **Wearable Sensor Challenges**: As an earring, it may need to be consistently worn or removed, reducing its effectiveness. **Limited Communication Range**: Utilises ZigBee with a 100 m range, which might not cover more significant properties or distances. **Battery Dependency**: The need for regular battery recharge or replacement could lead to lapses in monitoring.
Safety Turtle Pool Alarm [94], Water Patrol Child Guard [95]	Features a house-based base station and a wristband worn by children. The product is sealed and sturdy enough to endure children’s play. The wristband detects water immersion immediately and sends a radio signal to the base station, triggering a loud alarm. A single base station supports multiple wristbands. The alert system distinguishes between immersion and incidental contact with precipitation or sprinkler mist.	**Immersion Detection**: While immediate, it may not detect scenarios where a child is in danger without being fully immersed. **Wristband Design**: The effectiveness depends on the child consistently wearing the wristband. **Signal Interference**: Although it can ignore precipitation, other interference might affect the functionality of the alarm. **Single Base Station Limitation**: While it can connect to multiple wristbands, the single base station might have limited coverage for larger areas.

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
