# Peer review of "Enhancing Water Safety: Exploring Recent Technological Approaches for Drowning Detection"

_sensors, 2024, doi:10.3390/s24020331_

Round 1

Reviewer 1 Report

Comments and Suggestions for Authors

Reviewer Report

Thank you for the opportunity to review manuscript ID sensors-2679888 entitled ‘Enhancing Water Safety: Exploring Recent Technological Approaches for Drowning Detection’ which was submitted for potential publication in Sensors.

This manuscript aims to review the current evidence on drowning detection and surveillance systems.

General comments

Appropriate drowning terminology needs to be used throughout – near drowning is no longer accepted terminology. Non-fatal drowning is the term that must be used.

This review needs a method section or a clearer understanding of how these technologies and supporting literature were identified and the process for inclusion or exclusion, of course noting this is not a systematic review

Specific feedback

Introduction

Line 34 – replace traumatic with injury

Line 35 – accidental death should be accidental injury-related death

Suggest adding another reference to the end of the first paragraph to support the claims around age groups – see https://injuryprevention.bmj.com/content/26/Suppl_2/i83

Line 39 – reference needed for global evaluations vastly underestimate it – see here https://bmjopen.bmj.com/content/7/12/e019407 . Also by evaluation you mean estimation?

Line 40 – reference also needed for specific demographics at risk – recommend referencing this chapter - https://oxfordre.com/publichealth/display/10.1093/acrefore/9780190632366.001.0001/acrefore-9780190632366-e-307

Line 72 - the WHO indicates 90% of drowning occurs in low and middle income countries. Can you touch on affordability and accessibility of this technology and whether the approaches you document are restricted to use in high income countries? This needs to also be reflected in discussion and strengths and limitations of the work.

Section 2. drowning behaviours, signs and statistics

Im unsure why the focus on this single study from a HIC, US context. Would this not be better to reflect global statistics from WHO or the GBD study and/or reflect a diversity of countries and locations in this section? If you retain this example you need further commentary to identify the limitations of this approach and other risk factors that may not be present in this study yet contribute to the global drowning burden.

Line 103/04 – rephrase – spinal and brain injuries would be categorised elsewhere, not as drowning causes of death

Line 120 – near drowning is no longer accepted terminology since 2005 – see van beck et al – please remove and change to non-fatal drowning throughout paper.

Lack of referencing throughout this section, in particular the paragraph from 119-127

The entire paragraph from 128-135 needs to be deleted as no longer acceptable, again see van beck et al study

Line 137 – the organisation is royal life saving society Australia (RLSSA) please correct

Please update paragraph from 137 – with most recent 2023 drowning report

Table 1 seems to be missing video based sensors such as Poseidon system etc used in pools in Aus, US, France etc – there is literature already published on these systems and I believe they are mandatory under French law for public swimming pools – please update

Line 171 – needs a reference for economic impact of drowning see - https://www.sciencedirect.com/science/article/abs/pii/S0022437517306680

Line 465 – you can reference this study about importance of supervision. https://onlinelibrary.wiley.com/doi/abs/10.1111/jpc.14668

Line 472 – strongly suggest rephrasing ‘drain their family and society’ – this is really not acceptable language for the issue at hand

Section 4.2 – what are the limitations of these devices, has research on their effectiveness been done. A statement is also needed about being no absence for parental supervision

Lines 492-497 – these devices need referencing

Line 499 – unsure if the term most promising should be used – it is unclear how these approaches were identified and literature sourced, including any biases or gaps. Secondly, a more thorough discussion of the limitations or evaluation of these devices is needed.

Again I would like the authors to add some commentary around what can be taken from this review for LMICs were the vast majority of the drowning burden is?

Comments on the Quality of English Language

None ok aside from drowning terminology being outdated throughout and several strange phrases used which i have made suggestions on . 

Reviewer 2 Report

Comments and Suggestions for Authors

comments in the attachment

Reviewer 3 Report

Comments and Suggestions for Authors

Thank you for submitting the article.

It touches on a very difficult aspect related to water safety and minimizing the risk of drowning.

Please improve the readability of the article. Please put some spaces between important points, it will increase the readability of the article.

My conclusions lacked a clearly defined statement about which drowning detectiontion approaches is the best?

Regards

Comments on the Quality of English Language

The text contains minor linguistic and stylistic errors. Please read the text carefully again and correct any errors, or send it for language correction.

Round 2

Reviewer 1 Report

Comments and Suggestions for Authors

Thank you for the opportunity to review the revised manuscript ID sensors-2679888 entitled ‘Enhancing Water Safety: Exploring Recent Technological Approaches for Drowning Detection’ which was submitted for potential publication in Sensors.

Thank you to the authors for diligently addressing my concerns. The manuscript is much improved.

I would just like the authors to pay close attention to the reference list. Many of the references are incomplete – see refs 1, 5, 10, 17, 19, 20, 33, 81, 82, 87, 99, 100

These references all lack te necessary information for the reader to be able to locate them.

For example ref 5 should be referenced as follows:

Rahman, A., Peden, A. E., Ashraf, L., Ryan, D., Bhuiyan, A. A., & Beerman, S. (2021). Drowning: global burden, risk factors, and prevention strategies. In Oxford Research Encyclopedia of Global Public Health. https://doi.org/10.1093/acrefore/9780190632366.013.307

Duplicate reference – 23 and 31 are the same references

Comments on the Quality of English Language

Manuscript just needs a thorough proof for typos and flow